# RNA Sequencing Reveals a Strong Predominance of *THRA* Splicing Isoform 2 in the Developing and Adult Human Brain

**DOI:** 10.3390/ijms25189883

**Published:** 2024-09-13

**Authors:** Eugenio Graceffo, Robert Opitz, Matthias Megges, Heiko Krude, Markus Schuelke

**Affiliations:** 1Charité-Universitätsmedizin Berlin, Corporate Member of Freie Universität Berlin, Humboldt-Universität Berlin, and Berlin Institute of Health, Department of Neuropediatrics, 13353 Berlin, Germany; eugenio.graceffo@charite.de; 2Charité-Universitätsmedizin Berlin, Corporate Member of Freie Universität Berlin, Humboldt-Universität Berlin, and Berlin Institute of Health, Einstein Center for Neurosciences Berlin, 10117 Berlin, Germany; 3Charité-Universitätsmedizin Berlin, Corporate Member of Freie Universität Berlin, Humboldt-Universität Berlin, and Berlin Institute of Health, Institute of Experimental Pediatric Endocrinology, 13353 Berlin, Germany; robert.opitz@charite.de (R.O.); heiko.krude@charite.de (H.K.); 4Charité-Universitätsmedizin Berlin, Corporate Member of Freie Universität Berlin, Humboldt-Universität Berlin, and Berlin Institute of Health, Department of Pediatric Endocrinology, 13353 Berlin, Germany; matthias.megges@charite.de; 5Charité-Universitätsmedizin Berlin, Corporate Member of Freie Universität Berlin, Humboldt-Universität Berlin, and Berlin Institute of Health, Neurocure Clinical Research Center, 10117 Berlin, Germany

**Keywords:** thyroid hormone receptor, alternative splicing, organoids, single-cell RNA analysis, RNA sequencing, central nervous system

## Abstract

Thyroid hormone receptor alpha (THRα) is a nuclear hormone receptor that binds triiodothyronine (T3) and acts as an important transcription factor in development, metabolism, and reproduction. In mammals, THRα has two major splicing isoforms, THRα1 and THRα2. The better-characterized isoform, THRα1, is a transcriptional stimulator of genes involved in cell metabolism and growth. The less-well-characterized isoform, THRα2, lacks the ligand-binding domain (LBD) and is thought to act as an inhibitor of THRα1 activity. The ratio of THRα1 to THRα2 splicing isoforms is therefore critical for transcriptional regulation in different tissues and during development. However, the expression patterns of both isoforms have not been studied in healthy human tissues or in the developing brain. Given the lack of commercially available isoform-specific antibodies, we addressed this question by analyzing four bulk RNA-sequencing datasets and two scRNA-sequencing datasets to determine the RNA expression levels of human *THRA1* and *THRA2* transcripts in healthy adult tissues and in the developing brain. We demonstrate how 10X Chromium scRNA-seq datasets can be used to perform splicing-sensitive analyses of isoforms that differ at the 3′-end. In all datasets, we found a strong predominance of *THRA2* transcripts at all examined stages of human brain development and in the central nervous system of healthy human adults.

## 1. Introduction

Thyroid hormones (T4 and T3) are central to normal physiology throughout the animal kingdom, promoting growth, metabolism, reproduction, and, especially, brain development. In humans, this central role is underscored by the severe clinical problems of children born without thyroid hormone production—known as cretinism or congenital hypothyroidism—and by the almost completely normal development of these children when thyroid hormone treatment is initiated early after birth [1,2]. To exert their physiological effects, thyroid hormones must enter their many target cells via specific transporters (e.g., MCT8) to reach their two nuclear receptors, thyroid hormone receptor-alpha (THRα) and -beta (THRβ) [3]. While THRβ is primarily involved in the thyroid–pituitary feedback circuit to maintain normal serum thyroid hormone levels, THRα is expressed in a wide range of target tissues and, particularly, in the central nervous system.

THRα is a nuclear hormone receptor for the active hormone triiodothyronine (T3). T4 mainly acts as a prohormone to T3, although some recent evidence has shown weak binding to THRα, but to a much lesser extent than T3 [4]. THRα acts as a transcription factor for a variety of target genes involved in central nervous system development, metabolism, and growth [5]. In mammals, two major isoforms are produced by alternative splicing: THRα1 (encoded by *THRA1*), which contains a T3-sensitive ligand-binding domain (LBD), and THRα2 (encoded by *THRA2*), which lacks this T3-binding site and is therefore T3-independent (Figure 1a) [6]. In the absence of T3, THRα1 binds to the THR-responsive elements (TREs) located in the promoter regions of its target genes and represses gene expression [7]. Upon the binding of T3 to THRα1, conformational changes in the receptor promote coactivator binding and, ultimately, transcription of the target genes. In some instances, e.g., for the transcriptional regulation of thyrotropin-releasing hormone (*TRH*), the binding of T3 to THRα blocks *TRH* gene transcription at the hypothalamus [8].

In contrast to THRα1, THRα2 is unable to bind T3 and has been shown to act as a weak dominant-negative inhibitor of THRα1 action, possibly via protein–protein interactions [9]. In principle, due to these opposing effects, the molar ratio of THRα1 to THRα2 determines the response of the local target cell to T3; the more THRα1 present, the greater the T3 response and activation of the target gene that can be expected, and conversely, the more THRα2 present, the lower the target gene expression that can be expected (Figure 1b). However, the exact role and mechanisms of these opposing effects on human physiology are not well understood.

A recent study in mice demonstrated a predominance of the THRα1 isoform over the THRα2 isoform in cardiac and intestinal tissues and a predominance of the THRα2 isoform in neural tissues, suggesting a tissue-specific regulation and a physiological role for the THRα1 to THRα2 ratio [10].

To date, the functional relevance of these splice variants for the T3 response in human physiology has been demonstrated by disease-causing mutations in the coding sequences of the human *THRA* gene [11,12]. Most of these mutations abolish T3 binding to THRα1, thereby increasing the T3-unresponsive receptor pool of the target cell. This results in T3-resistance, with a severe phenotype mimicking congenital hypothyroidism, which is characterized by developmental delay, intellectual disability, growth deficit, bradycardia, delayed ossification, and an overall reduced metabolic rate (OMIM #614450) [11,12,13]. Despite such evidence for the importance of splicing in the pathophysiology of peripheral thyroid-hormone-resistance syndromes, the cellular *THRA1* to *THRA2* ratio has not been systematically investigated during human development or in healthy adults.

To address this knowledge gap, we collected and independently re-analyzed four (2 publicly available and 2 in-house generated) human bulk RNA-seq datasets and two publicly available human single-cell RNA-seq datasets generated from healthy adult tissues, brain cortical organoids, and human embryos during early development (Figure 1c). A schematic representation of the developmental stages covered by these datasets in relation to human development is shown in Figure 1d. We found widely varying absolute and relative abundances of the two splicing isoforms in different tissues and, similarly to previous studies in mice, a large overrepresentation of the *THRA2* splicing isoform in all neural tissues analyzed.

## 2. Results

### 2.1. Detection of THRA Isoforms in Bulk RNA-Seq Datasets

We analyzed the expression patterns of human *THRA* splicing isoforms in four different short-read bulk RNA-seq datasets: **D1**, an in-house generated dataset of 24 healthy pooled adult human tissues (GEO database: GSE224153); **D2**, a publicly available dataset of human cortical organoids from the laboratory of Giuseppe Testa (EMBL-EBI database: E-MTAB-8325) [14]; **D3**, an in-house generated dataset of T3-treated (GEO database: GSE250143) and untreated (GEO database: GSE250142) human cortical organoids (GEO database unifying code: GSE250144); **D4**, a publicly available dataset of human and gorilla cortical organoids from the laboratory of Madeline A. Lancaster (GEO database: GSE153076) [15] (Figure 1c; in light blue). We used StringTie to extract the transcripts per million (TPM) values of the two different splicing isoforms (Figure 1e; light blue). The TPM values of *THRA1* and *THRA2* in all the bulk RNA-seq datasets are shown in Figure 2, Figure 3, Figure 4 and Figure 5. In adult tissues, all the nervous system samples expressed high levels of total *THRA* (Figure 2; dataset D1, >100 TPM, full length of the bar). In addition, all the nervous system tissues expressed higher levels of *THRA2* compared with *THRA1* (>70% of total *THRA*; dark blue in Figure 2). In other non-neural tissues, with the exception of smooth muscle, *THRA* expression was lower than in neural tissues, with a clear predominance of *THRA2* in the thyroid, kidney adrenal gland, heart, and pancreas, whereas the *THRA1* isoform (samples below the red line, >50% of total THRA; light blue in Figure 2) was more prevalent in adipose tissue, skeletal muscle, the lung, and the colon. In order to look for a potential functional relevance of different *THRA1*:*THRA2* ratios, we provided the TPM values of the well-known THRα target genes *IGF1* [16,17], *DIO3* [18], and *RXRA* in Appendix A, where we see a tendency for higher expression in tissues with a comparatively lower *THRA2* expression. However, as a caveat, the levels of thyroid hormone receptor (THR)-regulated genes in different tissues are likely to be determined by many factors other than the splicing ratio of *THRA* isoforms. The final expression levels of target genes would therefore depend on factors such as the amount of T3 generated in these cells (e.g., by the various deiodinases), the co-presence of *THRB* and, importantly, on epigenomic factors that dictate promoter accessibility for any given THR-regulated gene. In addition, many THR-regulated genes are tissue-specific in the first place, which would make direct comparisons very challenging.

In all further analyses, we focused on neural tissues with cortical organoid datasets, which allowed us to study developmental changes in *THRA* isoform expression. In all the human organoid datasets, we detected an increase in total *THRA* mRNA expression over time as well as a strong predominance of *THRA2* over *THRA1* at all time points analyzed (>80% of total *THRA*; dark blue in Figure 3a and Figure 4a). Notably, this pattern was fully conserved in gorilla cerebral organoids (dataset **D4**; Figure 5). We also examined whether acute 48 h treatment of human cortical organoids with high concentrations of T3 (50 nM) affected *THRA* isoform expression but found no treatment effects on expression levels of total *THRA* mRNA or on specific splicing isoforms (Figure 4b–d; one-way ANOVA, *p* > 0.05).

### 2.2. Detection of THRA Splicing Isoforms in Single-Cell RNA-Seq Datasets

To study *THRA* isoform expression levels in the developing brain at a cellular resolution, we mined publicly available single-cell RNA (scRNA)-seq databases. To date, there is a lack of publicly available scRNA-seq datasets of human cortical organoids or human embryos generated by a splicing-isoform-sensitive platform (e.g., Smart-Seq2 and long-read sequencing). In contrast, short-read 10X Chromium datasets are widely available, but these tend to have an intrinsic bias toward either the 3′- or 5′-ends of their mRNA preparations depending on the specific protocol used, and, thus, are generally not suitable for isoform-specific analyses. However, the fact that *THRA1* and *THRA2* differ precisely at their 3′-ends provided the justification to re-edit the genome reference to represent *THRA1* exclusively by exon 9b + its 3′UTR and *THRA2* exclusively by exon 10 + its 3′UTR (Figure 6a; see the Methods section for more details). We validated whether these exons could be used as a proxy to investigate the predominance of either *THRA1* or *THRA2* in the bulk RNA-seq dataset **D2**. The relative abundances of *THRA1* and *THRA2* detected by the exon proxy (Figure 3b) are comparable to those detected by StringTie (Figure 3a). Next, we downloaded and bioinformatically reanalyzed two scRNA-seq datasets generated on the 10X Chromium platform: **D5**, a dataset of human cortical organoids from the laboratory of Giuseppe Testa, collected at days 50 and 100 after neural induction (EMBL-EBI database: E-MTAB-8337 [9]); and **D6**, a dataset of whole healthy human embryos from Carnegie stages 12 to 16 (corresponding to post conception days 30–39) from the laboratories of Zhirong Bao and Weiyang Shi (GEO database: GSE157329 [19]; Figure 1c in beige). The original authors had processed the embryos by cutting them and preparing the separate sequencing libraries of each segment. We downloaded FastQ files of the samples annotated by the original authors as either “head_trunk” or “head”.

Cell Ranger v7.1.0 was used to map FastQ files to the custom (GRCh38) human reference genome. We used the Seurat package for data analysis, e.g., to filter low quality cells, normalize counts, integrate different time points, identify clusters, and visualize results (Figure 1e, beige).

Figure 6b shows the uniform manifold approximation and projection (UMAP) representation of the **D5** dataset. The identified cell populations recapitulate different stages of cortical development. This includes the progenitor pool of radial glia (marker: *VIM*) and outer radial glia (markers: *FABP7*, *PTN*), a pool of cells that were positive for neuronal differentiation markers (*GAP43*, *DCX*, *SYT1*, *STMN2*), and, finally, pools of cells that were positive for markers of early oligodendrogenesis (*PLP1*) and early astrocytogenesis (*APOE*).

In addition, we identified TTR-positive choroid plexus cells, as did the original authors of the dataset. The expression levels of the marker genes for each cell cluster are shown in Figure 6c. Expression of *THRA1* and *THRA2* isoforms could be detected across the different cell clusters, whereas we observed a strong predominance of *THRA2* over *THRA1* in all cell types (Figure 6d). Furthermore, when comparing THRA2 expression levels between cell populations, we detected an increased transcript abundance in more differentiated cell populations compared with immature progenitors. This pattern was confirmed when samples were grouped by day 50 and day 100 time points (Figure 6e). Interestingly, when comparing expression levels in the progenitor populations (i.e., radial glial cells, ribosome-enriched radial glial cells, and outer radial glia) between day 50 and day 100, we observed higher expression levels of *THRA2* at day 100, indicating that even progenitor cells increase their expression of *THRA2* over time.

Figure 7a shows a UMAP representation of the **D6** dataset with the cell labels provided by the authors of the dataset. To be able to compare the results with dataset **D5**, we sub-clustered and further analyzed the cell populations originally labeled as “Neuron”, “Neural progenitor”, “Sensory neuron”, and “Schwann cell”. Figure 7b shows the expression levels of cell markers for each population, namely *DCX*, *GAP43*, and *STMN2* for neurons; *POU4F1* for sensory neurons; *PLP1*, *MPZ*, and *SOX10* for Schwann cells; and *PAX6*, *SOX2*, and *VIM* for neural progenitors. We observed a strong predominance of *THRA2* over *THRA1* in all cell types and a higher expression of *THRA2* in differentiated neurons and Schwann cells compared with neural progenitors and sensory neurons (Figure 7c).

## 3. Discussion

In this study, we independently analyzed four bulk RNA-seq and two single-cell RNA-seq datasets to tease out the expression patterns of the major THRA splicing isoforms (*THRA1* vs. *THRA2*) in healthy adult human tissues and in cortical organoids. We provide an example of how single-cell RNA-seq datasets using the 10X Chromium platform and short-read sequencing, generally thought to be unsuitable for isoform detection, can actually be adapted to distinguish specific isoforms that differ at their 3′-ends by editing the underlying gene annotation file. Based on these datasets, we describe a strong predominance of *THRA2* transcripts at all stages of early human brain development as well as in the central nervous system of healthy adult human tissue. Although first described more than 30 years ago, the role of THRα2 remains largely unknown [6,20,21]. Several studies have described its weak dominant-negative effect (DNE) on THRα1-mediated gene expression [22,23,24,25,26]. Authors have suggested that the competitive binding to TREs may be the mechanism of action, but recent findings indicated a lower affinity for TREs by THRα2 compared with THRα1 [27]. Therefore, the inhibitory effect of THRα2 may also be mediated upstream of receptor binding through its competitive binding of cofactors and by the formation of inactive heterodimers [27].

Thus, THRα2 is thought to play a role in modulating the effects of T3 on cellular growth and homeostasis by antagonizing THRα1-mediated effects of T3 in a spatiotemporal manner, a mechanism that occurred only in the evolution of mammals in eutherians after the divergence from marsupials and monotremata [28].

Our results of a strong predominance of THRα2 in the central nervous system extend this idea and even suggest a protective role of THRα2 against thyroid-hormone-regulated gene expression, at least in the developing brain. This T3-antagonizing role of THRα2 may have gained importance during mammalian evolution as the fetus in eutherians comes under the influence of high maternal T3 concentrations within its complex placenta.

Whether the detected transcript levels translate into comparable protein levels requires further confirmation. However, given the strong expression of THRα2 at the RNA level, one could hypothesize that the *THRA2* transcript may also act as a long non-coding RNA in addition to being a template for protein synthesis. Some examples of such dual-function RNA transcripts have been described in the context of cancer [29,30]. In addition, considering that exon 10 of *THRA2* is antisense to *NR1D1*, it may down-regulate the expression levels of the resulting Rev-Erbɑ protein, which is involved in circadian metabolic control [31].

In addition, both the expression and the activity of *THRA* isoforms are regulated post-transcriptionally and post-translationally. In particular, dephosphorylation of THRα2 has been shown to increase the DNA-binding affinity and, thus, the inhibitory function of THRα2 [32], and sumoylation has been shown to affect its interaction with other TRs and, thus, gene expression [33,34].

Taken together, our findings with the detected difference in THRα1 vs. THRα2 expression should be kept in mind when inferring biological mechanisms from molecular studies that do not distinguish between the two isoforms. Further expression studies should examine the isoform expression pattern at later stages of fetal brain development and at postnatal life stages. In addition, the tight tissue- and time-dependent regulation of THRα splicing suggests the presence of additional genetic alterations in the non-coding sequences of the THRα locus that, by disrupting the balance of THRα isoforms, may cause additional as-yet undiagnosed inherited diseases associated with the THRα gene.

Finally, open science, data sharing, and reuse have recently advanced scientific progress by promoting collaboration, transparency, and cost-effectiveness. A large number of scRNA-seq datasets generated on the 10X Chromium platform are available for researchers to use. The 10X Chromium platform is known for its high throughput and scalability and is widely used [35]. However, these datasets are generally not suitable for isoform-sensitive analyses due to the short-read sequencing processing and the 3′- or 5′-end bias of the library preparation. Given the unavailability of open-access single-cell datasets of human cortical organoids generated by isoform-sensitive platforms such as Smart-seq2 and long-read sequencing, we took advantage of the fact that *THRA1* and *THRA2* differ precisely at their 3′-ends and edited a reference genome to map *THRA1* and *THRA2* to their specific exons. Despite the relatively shallow sequencing depth of the 10X Chromium platform, both isoforms were detected, and the difference in UMI counts between the two isoforms was large enough to confirm the predominance of *THRA2* in all cell populations. This study provides an example of how widely available scRNA-seq datasets generated with the 10X Chromium platform can be used to perform isoform-sensitive analyses in specific scenarios.

## 4. Methods

### 4.1. Healthy Adult Human Tissues—Dataset D1

Dataset **D1** includes 10 different nervous system tissues and 14 non-neural tissues, with each sample pooled from an average of six healthy individuals (range: 1–21 individuals, male/female, Asian, Caucasian, and African American, aged 18–89 years), which does not allow for the identification of any contributor. Cause of death was reported as sudden death or traffic accident. High-quality total RNA was purchased from Takara (www.takarabio.com, accessed on 7 September 2024). Total RNA was isolated by the guanidinium thiocyanate method [36]. Total RNA concentration, integrity, and purity were analyzed by capillary electrophoresis (CE) using the Agilent 2100 Bioanalyzer and the Agilent Fragment Analyzer (Agilent Technologies, Santa Clara, CA, USA). A detailed overview of the dataset is provided in Appendix A. As this dataset was generated from anonymous pooled commercially purchased RNA samples that do not allow for the identification of individual patients, no ethical or protocol approval was required. The rules of good scientific practice by the Charité Medical School were followed.

### 4.2. Human Cortical Organoids—Dataset D3

Cortical organoids were generated as described in a previous publication [37]. All hiPSC lines used in this study were obtained from the hiPSC biobank at the Berlin Institute of Health (Berlin, Germany). Information on donors, cell line derivation, ethical approval, and third-party availability for all lines is available at the Human Pluripotent Stem Cell Registry (https://hpscreg.eu/, accessed on 7 September 2024).

Briefly, cerebral organoids were derived from hiPSCs of n = 3 healthy donors. The BIHi001-B (male), BIHi043-A (female), and BIHi005-A (male) lines were cultured in 6-well plates coated with hESC-qualified Geltrex (Thermo Fisher, #A1413302) in E8 medium (Thermo Fisher, #A1517001) under hypoxic conditions. Media were changed daily. hiPSC cultures were split at 80% confluence by incubation in 0.5 mM EDTA diluted in PBS (5 min at 37 °C) followed by resuspension of cell clumps in 1.5 mL E8 medium. Cells were seeded at a split ratio of 1:20 or 1:30. Organoids were grown using the STEMdiff™ Cerebral Organoid Kit (StemCell Technologies, #08570, #08571) according to the manufacturer’s instructions and collected on days 21, 41, and 61. A subset of organoids was treated with 50 nM T3 for 48 h prior to tissue collection. Samples were snap-frozen in liquid nitrogen and stored at −80 °C until RNA extraction. RNA was extracted using the NucleoSpin RNA Plus Kit from Macherey and Nagel (#740984) according to the manufacturer’s guidelines. A detailed overview of the dataset is provided in Appendix A.

The source cells (fibroblasts) for the generation of the above iPSCs were purchased from the ATCC, which had obtained ethical consent from the cell donors. Details of the consent(s) are listed on the Human Pluripotent Stem Cell Registry homepage for each cell line.

### 4.3. Bulk RNA Sequencing—Datasets D1 and D3

At least 500 ng of each sample from the **D1** and **D3** datasets were sent out for bulk RNA sequencing, which was performed at the Beijing Genomics Institute (BGI, Shenzhen/Hong Kong, China, https://www.genomics.cn, accessed on 7 September 2024) using their DNBseq platform on a strand-specific cDNA library, generating 60 million 100 bp paired-end reads. The library preparation included an mRNA enrichment step using oligo(dT)-attached magnetic beads.

### 4.4. Publicly Available Datasets—D2, D4, D5, and D6

The FASTQ files of the **D2**, **D4**, **D5**, and **D6** datasets were downloaded from either the Gene Expression Omnibus (GEO) database (**D4** GSE153076, **D6** GSE157329) or the EMBL-EBI database (**D2** E-MTAB-8325, **D5** E-MTAB-8337). Specifically, for dataset **D2**, we used the paired-end reads from all the untreated samples used as controls by the original authors (n = 3 replicates each from days 18, 50, and 100 post-neuronal induction). For dataset **D4**, we used all the available samples (n = 3 replicates each from days 0, 2, 3, 5, 10, 15, and 25 post-neuronal induction for human cortical organoids and n = 3 replicates from days 0, 2, 3, 5, 10, 15, and 25 for gorilla cortical organoids). For dataset **D5**, we downloaded all the untreated samples that were used as controls by the authors (n = 3 replicates each from days 50 and 100 post-neuronal induction). For dataset **D6**, we used all the samples containing the head sections of the embryos, namely “embryo1-head_trunk”, “embryo2-head”, “Emn.04-head”, “Emb.05-head”, “Emb.06-head a”, and “Emb.06-head b” (n = 72 FASTQ files). Details of all samples included in the analysis are summarized in Appendix A.

### 4.5. Bioinformatic Analyses of Bulk RNA-Seq Datasets D1, D2, D3, and D4

Datasets **D1**, **D2**, **D3**, and **D4** were analyzed using the same pipeline (Figure 1e; light blue). Sequence quality was assessed using FastQC v0.11.8 (www.bioinformatics.babraham.ac.uk/projects/fastqc, accessed on 7 September 2024) and MultiQC v1.6 [38]. Reads were mapped to the EMBL human genome 38, patch release 13 (**D1**, **D2**, **D3**, and **D4** human samples) or to the modified UCSC Kamilah gorilla genome GGO, version 6 (**D4** gorilla samples; see below for details) using the splice-aware aligner STAR v2.7.10a [39]. BAM files were sorted and indexed using SAMtools v1.9 [40]. Mapping quality was assessed by calculating junction_saturation.py, inner_distance.py, read_distribution.py, read_duplication.py with RSeQC v2.6.4 [41]. StringTie v2.1.7 was used to extract the transcripts per million (TPM) values of all the known *THRA* variants in each tissue [42]. For *THRA1* expression, we summed up the TPM values of the variants containing exon 9b (human samples: ENST00000450525 and ENST0000054624; gorilla samples: XM_019026893.2). For *THRA2* expression, we summed up the TPM values of the variants containing exon 10 (human samples: ENST00000264637 and ENST00000584985; gorilla samples: custom *THRA2*; see below for details). FeatureCounts v2.0.1 was used to extract read counts for exon 9b [chr17:40,089,334-40,092,627(+)] and exon 10 [chr17:40,093,020-40,093,867(+)] for proxy validation [43]. Data visualization and statistical analyses were performed in R v4.0.0 (https://www.r-project.org/, accessed on 7 September 2024).

### 4.6. Generation of a Custom Gorilla Genome Reference with the THRA2 Locus

To date, the mRNA transcript of the gorilla *THRA2* isoform has not been deposited in the NCBI Reference Transcriptome. To localize the genomic coordinates of *THRA* exon 10 in the gorilla genome, we manually BLAST-aligned the human *THRA* exon 10 sequence to the gorilla genome [chr5:41,857,943-41,858,305(-)]. We then created a custom reference genome by adding an entry for *THRA2* [transcript chr5:41,857,944-41,888,473(-); exons 1 to 9 same as *THRA1* entry; exon 10 chr5:41,857,944-41,858,305(-); 3′UTR chr5:41,857,457-41,857,944(-); stop codon chr5:41,857,941-41,857,943(-)]. The modified annotation of the gorilla reference genome has been uploaded to figshare (https://doi.org/10.6084/m9.figshare.24486601, accessed on 7 September 2024).

### 4.7. Bioinformatic Analyses of Single-Cell RNA-Seq Datasets D5 and D6

Datasets **D5** and **D6** were analyzed individually using a similar pipeline (Figure 1e; beige). Sequencing quality was assessed using FastQC v0.11.8 and MultiQC v1.6. Cellranger v7.1.0 was used to map the reads to our custom human reference genome (see below for details) and to extract count matrices [44]. We followed a standard workflow for data preprocessing using the Seurat package v4.3.0 [45] in R v4.0.0.

Quality-control preprocessing included filtering out low-quality cells with mitochondrial gene counts greater than 4–10% and cells with feature counts less than 200 or 1000 and greater than 9000 or the 90th centile. Table 1 shows the quality-control metrics for each sample (i.e., number of cells before filtering, median number of genes per cell, % mitochondrial gene threshold, feature count threshold, and number of cells after filtering). Data were normalized and scaled (regressing out % mitochondrial genes), and highly variable genes were calculated using the SCTransform() function of the Seurat package.

For both datasets, samples corresponding to different time points and replicates were integrated using the Seurat package and the CCA Pearson residual method. Specifically, we sequentially ran Seurat’s SelectIntegrationFeatures(), PrepSCTIntegration(), FindIntegrationAnchors(), and IntegrateData() with 3000 integration features.

For dataset **D5**, we identified clusters using the FindClusters() function (resolution = 0.5) and manually annotated cell types based on the original authors’ annotations and cell -type-specific markers found through multiple rounds of sub-clustering. For dataset **D6**, we used the annotations from the original publication and sub-clustered and further analyzed the cell populations originally labeled as “Neuron”, “Neural progenitor”, “Sensory neuron”, and “Schwann cell”.

### 4.8. Generation of a Customized Human Genome Reference for the Analysis of the scRNA-Seq Datasets

We edited the genome reference to map exon 9b and the corresponding UTR, stop codon, and CDS exclusively to *THRA1* and exon 10 exclusively to *THRA2* (Figure 6a). To achieve this, entries corresponding to exon 10 [chr17:40,093,020-40,093,613(+), chr17:40,093,137-40,093,867(+) and chr17:40,093,020-40,093,867(+)] were given a new gene_id, transcript_id, and the gene_name “*THRA2*” to make it count as a separate gene and to allow them to be mapped with Cellranger. Entries corresponding to exon 9b were given a new gene_id, transcript_id, and the gene_name “*THRA1*”, and their start position was edited to avoid overlapping with exon 9a, which maps to *THRA* [chr17:40,089,334-40,092,627(+) and chr17:40,089,334-40,089,730(+)]. Exons 1 to exon 9a were set to map to *THRA*. The generated custom human reference genome has been uploaded to figshare (https://doi.org/10.6084/m9.figshare.24486535.v2, accessed on 7 September 2024).

## 5. Code Availability

The R codes used to analyze both bulk and single-cell datasets are available at https://github.com/eugeniograceffo/Graceffo_et_al_2024_IJMS (accessed at on 7 September 2024).

Mapping of the bulk RNA-seq FASTQ files was performed with the following STAR options:STAR --runThreadN 60 \--readFilesCommand gunzip -c \--twopassMode Basic \--outFilterIntronMotifs RemoveNoncanonical \--outSAMstrandField intronMotif \--outSAMtype BAM Unsorted \--alignSoftClipAtReferenceEnds No \--outFilterMatchNmin 100 \--outSAMattrIHstart 0 \--alignEndsType EndToEnd

StringTie was run as follows:stringtie -o ${file%.sorted.bam}.stringtie_output.gtf \-e \--rf \-p 60 \-A ${file%.sorted.bam}.gene_abund.tab \-C ${file%.sorted.bam}.cov_refs.gtf

## Figures and Tables

**Figure 1 ijms-25-09883-f001:**
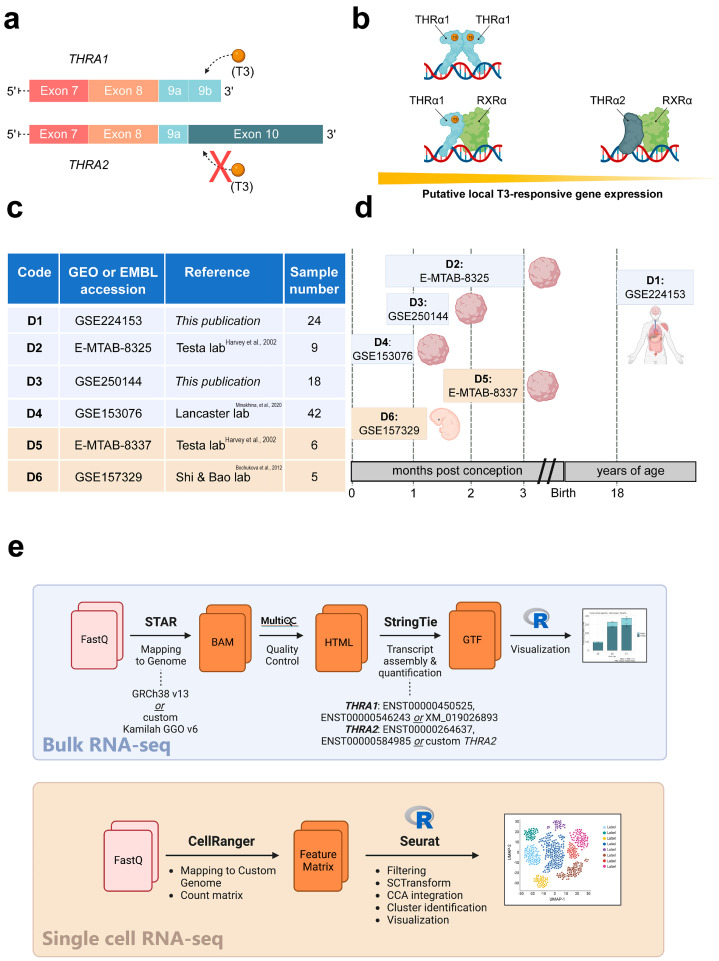
**Study overview.** (**a**) Schematic representation of the 3′-ends of the *THRA* isoform 1 and 2 mRNAs encoding THR*α*1 and THR*α*2, respectively. The orange spheres represent the T3 ligand, and the solid rectangles represent the exons. The schematic highlights that T3 can bind to THR*α*1 but not to THR*α*2. (**b**) Schematic representation of local T3-responsive gene expression based on the abundance of THR*α*1 and THR*α*2. In the presence of the same amount of local T3, cell types that synthesize more THR*α*1 will have greater T3-responsive gene expression compared with cell types that synthesize more THR*α*2. (**c**) Overview of the datasets used in this study. Datasets with a **blue** background are bulk RNA-seq datasets, while datasets with a **beige** background are single-cell RNA-seq datasets, [9,10,11]. (**d**) Schematic representation of the developmental stages covered by the datasets in relation to human development. Datasets with a **blue** background are bulk RNA-seq datasets, while datasets with a **beige** background are single-cell RNA-seq datasets. (**e**) Bioinformatic pipelines used to analyze the **bulk RNA-seq datasets** and the **single-cell RNA-seq datasets**.

**Figure 2 ijms-25-09883-f002:**
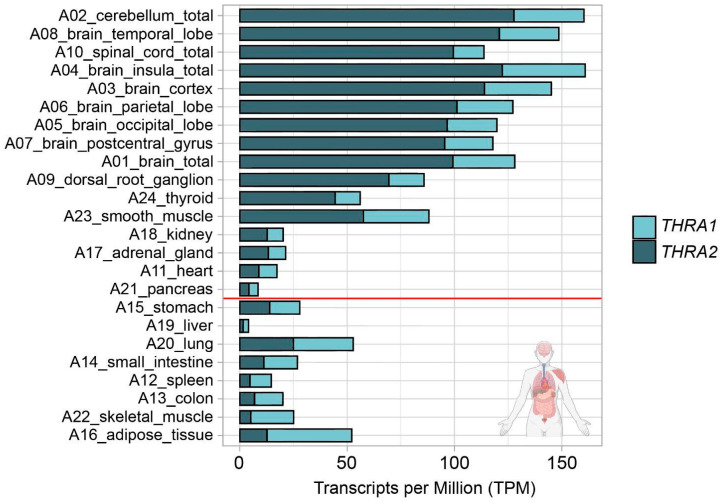
***THRA* isoform expression pattern of the D1 GSE224153 dataset in transcripts per million (TPM)**. The graph shows that all the nervous system samples expressed high levels of total *THRA* (full bar length), with a predominance of the *THRA2* isoform (dark blue). Samples are sorted in decreasing order based on the difference between *THRA2* and *THRA1* (light blue). Samples above the red line express more *THRA2* than *THRA1*. Data from ~62 M uniquely mapped reads.

**Figure 3 ijms-25-09883-f003:**
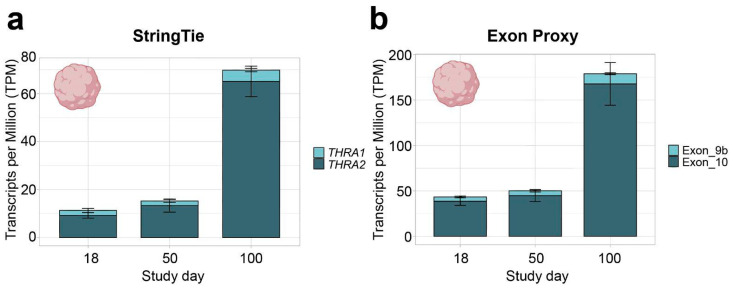
***THRA* isoform expression pattern of the D2 E-MTAB-8325 dataset in transcripts per million (TPM)**. (**a**) TPM calculated with StringTie (full transcript length) showing an increase in total *THRA* expression (full bar height) over time and a strong predominance of *THRA2* (dark blue) at all time points. Bar plots show the mean ± SEM; n = 3. (**b**) TPM calculated using exon 9b as a proxy for *THRA1* (light blue) and exon 10 as a proxy for *THRA2* (dark blue). The increased TPM values in (**b**) compared with (**a**) agree with the fact that TPM calculations normalize against transcript length; hence, using the shorter exon instead of the full transcript results in higher TPM values. The plots show a similar expression pattern as in (**a**), indicating that exons 9b and 10 can be used as proxies for isoform expression. Bar plots show the mean ± SEM; n = 3. Data from ~51 M uniquely mapped reads.

**Figure 4 ijms-25-09883-f004:**
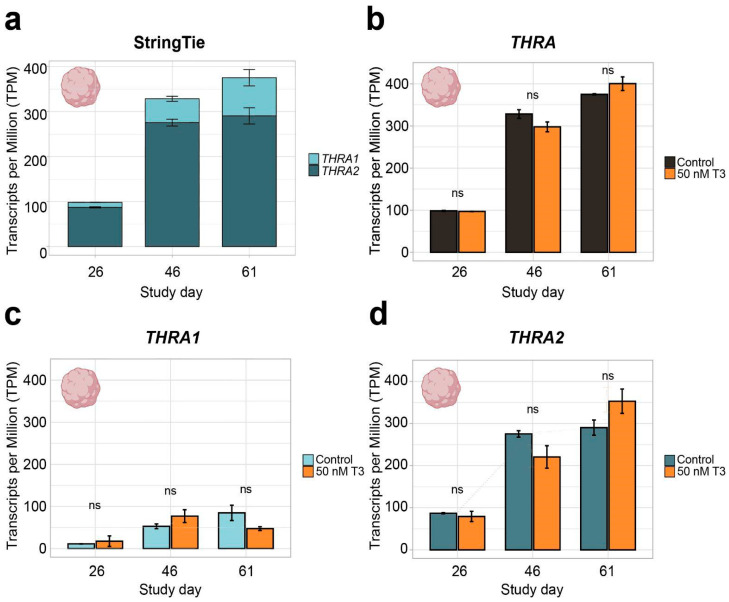
***THRA* isoform expression pattern of dataset D3 GSE250144 in transcripts per million (TPM)**. (**a**) Graph showing an increase in total *THRA* (full bar length) expression over time and a strong predominance of *THRA2* (dark blue) over *THRA1* (light blue) at all time points. Bar plots show the mean ± SEM; n ≥ 2. (**b**) Expression levels of total *THRA* (black) showing no difference between controls and samples treated with 50 nM T3 for 48 h before collection (orange). ns = not significant; mean ± SEM; one-way ANOVA test; n ≥ 2. (**c**) Expression levels of *THRA1* (light blue) showing no difference between the controls and samples treated with 50 nM T3 for 48 h before collection (orange). ns = not significant; error bars represent the mean ± SEM; one-way ANOVA test; n ≥ 2. (**d**) Expression levels of *THRA2* (dark blue) showing no difference between the controls and samples treated with 50 nM T3 for 48 h before collection (orange). ns = not significant; error bars represent the mean ± SEM; one-way ANOVA test; n ≥ 2; data from ~78 M uniquely mapped reads.

**Figure 5 ijms-25-09883-f005:**
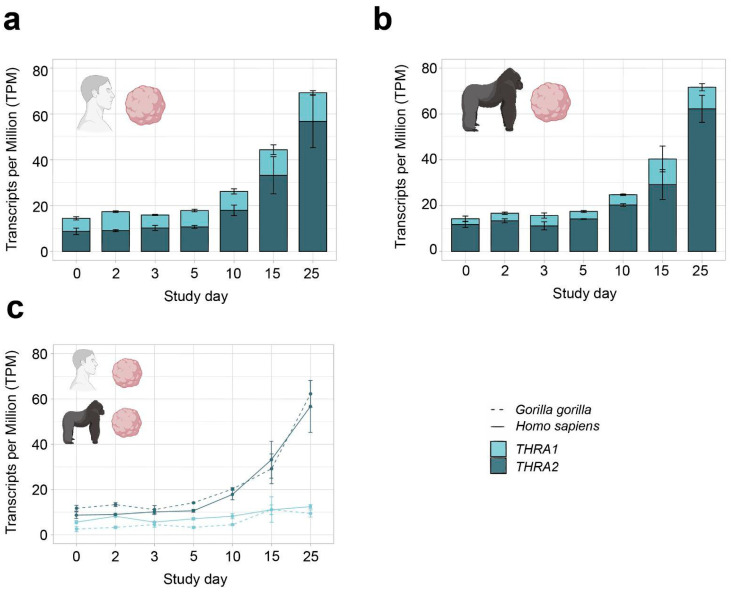
***THRA* isoform expression pattern of the D4 GSE153076 dataset in transcripts per million (TPM)**. (**a**) Graph showing an increase in total *THRA* expression (full bar length) over time and a strong predominance of *THRA2* (dark blue) over *THRA1* (light blue) at all time points in *Homo sapiens* samples. Bar plots show the mean ± SEM; n = 3. (**b**) Graph showing an increase in total *THRA* expression (full bar length) over time and a strong predominance of *THRA2* (dark blue) over *THRA1* (light blue) at all time points in *Gorilla gorilla* samples. Bar plots show the mean ± SEM; n = 3. (**c**) Graph comparing the expression levels of *THRA1* (light blue) and *THRA2* (dark blue) in *Homo sapiens* (solid line) vs. *Gorilla gorilla* (dashed line); error bars show the mean ± SEM; n = 3. Data from ~20 M uniquely mapped reads.

**Figure 6 ijms-25-09883-f006:**
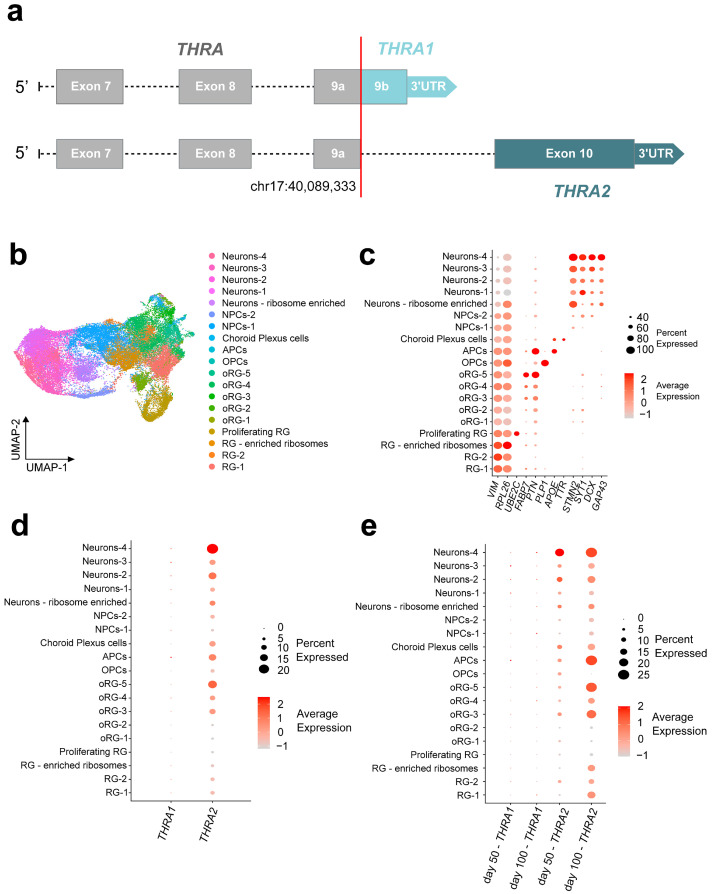
***THRA* isoform expression pattern of human cortical organoids from a single-cell dataset (D5 E-MTAB-8337)**. (**a**) Schematic of the edits made to the human reference genome annotation to detect *THRA* isoforms in 10X Chromium scRNA-seq datasets. Exons up to exon 9a [chr17:40,089,33] were considered *THRA* (in **gray**), exon 9b and its 3′-UTR were specifically mapped to *THRA1* (light blue), and exon 10 and its 3′-UTR were specifically mapped to *THRA2* (dark blue). (**b**) UMAP plot showing the cell types identified by manual curation. These include a group of (i) progenitor radial glial cells (RG-1, RG-2, RG-enriched ribosome and proliferating RG), (ii) outer radial glial cells (oRG-1 to oRG-5), (iii) intermediate neuronal precursor cells (NPCs-1 and NPCs-2), (iv) early differentiated neurons (Neurons-1 to Neurons-4 and Neurons—ribosome enriched), (v) early astrocyte precursor cells (APCs), (vi) early oligodendrocyte precursor cells (OPCs), and, finally, (vii) choroid plexus TTR^+^ cells (choroid plexus cells). (**c**) Dot plot showing the relative expression levels of the gene markers used to identify each population. Markers: *VIM* for radial glia, *RPL26* for ribosome enrichment, *UBE2C* for proliferating radial glia; *FABP7* and *PTN* for outer radial glia; *PLP1* for oligodendrocytic lineage; *APOE* for astrocytic lineage; *TTR* for choroid plexus cells; and *STMN2*, *SYT1*, *DCX*, and *GAP43* for neurons. (**d**) Dot plot showing the relative expression levels of *THRA1* and *THRA2* isoforms in each cell population (all n = 7 samples and time points combined). (**e**) Dot plot showing the relative expression levels of *THRA1* and *THRA2* isoforms in each cell population separated by the time point of analysis. NPC, neural precursor cell; APC, astrocyte precursor cell; OPC, oligodendrocyte precursor cell; oRG, outer radial glia; RG, radial glia; UMAP, Uniform Manifold Approximation and Projection.

**Figure 7 ijms-25-09883-f007:**
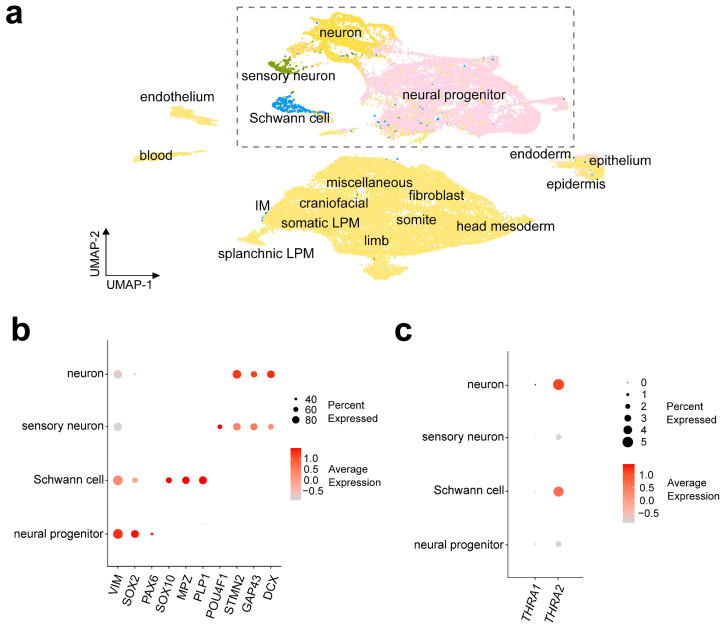
***THRA* isoform expression pattern of a human embryo single-cell dataset (D6 GSE157329)**. (**a**) UMAP plot showing the cell types identified by the authors of the dataset. The dashed rectangle highlights the four cell populations that were sub-clustered and used for further downstream analysis. (**b**) Dot plot showing the relative expression levels of gene markers in the identified cell populations. Markers: *VIM*, *SOX2*, and *PAX6* for neural progenitors; *SOX10*, *MPZ*, and *PLP1* for Schwann cells; *POU4F1* for sensory neurons; and *STMN2*, *GAP43*, and *DCX* for neurons. (**c**) Dot plot showing the relative expression levels of *THRA1* and *THRA2* isoforms in each cell population (data from all n = 6 embryos of Carnegie stages 12–16 combined). UMAP, uniform manifold approximation and projection.

**Table 1 ijms-25-09883-t001:** Filtering thresholds used to analyze each single-cell RNA-seq dataset.

Dataset Code	Origin	GEO or EMBL Accession Numbers	Submitted File Name	Study Day	Number of Cells before Filtering	Median Genes per Cell	Threshold % of mtDNA Genes	Threshold Number of Features [Quantile]	Number of Cells after Filtering
**D5**	Testa lab	E-MTAB-8337	3391B_D50	50	5576	2566	<5%	>200 and <9000	5558
**D5**	Testa lab	E-MTAB-8337	kolf2c1day50	50	11,785	2576	<4%	>200 and <9000	11,739
**D5**	Testa lab	E-MTAB-8337	G1_MIF1D50	50	4602	1701	<6%	>200 and <9000	4577
**D5**	Testa lab	E-MTAB-8337	S20271_156	100	2345	2379	<8%	>200 and <9000	2251
**D5**	Testa lab	E-MTAB-8337	S20269_154	100	1389	1853	<9%	>200 and <9000	1356
**D5**	Testa lab	E-MTAB-8337	S20265_136	100	9610	1754	<10%	>200 and <9000	9562
**D5**	Testa lab	E-MTAB-8337	S20276_177	100	3188	1619	<5%	>200 and <9000	3168
**D6**	Shi and Bao lab	GSE157329	embryo1-head_trunk	CS 12	21,570	2879	<10%	>1000 and <[90th]	13,462
**D6**	Shi and Bao lab	GSE157329	embryo2-head	CS 13–14	13,193	1546	<10%	>1000 and <[90th]	4386
**D6**	Shi and Bao lab	GSE157329	Emb.04-head	CS 13–14	18,249	2321	<10%	>1000 and <[90th]	8734
**D6**	Shi and Bao lab	GSE157329	Emb.05-head	CS 15–16	17,246	2894	<10%	>1000 and <[90th]	12,926
**D6**	Shi and Bao lab	GSE157329	Emb.06-head a	CS 15–16	8109	2382	<10%	>1000 and <[90th]	3406
**D6**	Shi and Bao lab	GSE157329	Emb.06-head b	CS 15–16	8357	1942	<10%	>1000 and <[90th]	3426

## Data Availability

The paired-end FASTQ files of the **D1** dataset have been submitted to the GEne Omnibus (GEO) database under the accession number GSE224153. RNA sample metadata, e.g., Takara lot numbers, and quality statistics are available in the accompanying metadata spreadsheet of the GSE224153 database entry and in Appendix A. The FASTQ files of the **D3** dataset have been submitted to the GEO database under the accession number GSE250142 (native cortical organoids) and GSE250143 (T3-treated cortical organoids), and the metadata are available in Appendix A (datasets GSE250142 and GSE250143 are combined into dataset GSE250144). Datasets **D2**, **D4**, **D5**, and **D6** are publicly available under the accession numbers given above.

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
