# Peer review of "RNA Sequencing Reveals a Strong Predominance of THRA Splicing Isoform 2 in the Developing and Adult Human Brain"

_ijms, 2024, doi:10.3390/ijms25189883_

Round 1

Reviewer 1 Report

Comments and Suggestions for Authors

In this work Graceffo et al have mined two novel RNAseq datasets generated by the authors, as well as publicly available bulk RNAseq and single cell RNAseq databases, to determine the ratio of THRA1 and THRA2 expression in several tissues, with a special focus on the central nervous system. While THRA1 encodes the thyroid hormone receptor THRα1, THRA2 encodes THRα2, which lacks the ability of binding T3 and is believed to act as and inhibitor of THRα1 activity. Even though the biological significance of these findings is yet to be determined, they provide interesting reflexions on how specific tissues and  specific cell types might regulate their sensitivity to T3 by increasing THRA2 expression throughout development. The research has been thoroughly conducted, and the manuscript is well-written and easy to follow.

I have minor queries aimed at improving clarity:

-          Line 44: When talking about thyroid hormone replacement treatment shortly after birth to prevent neurological impairments, I strongly recommend also citing Gabriella Morreale’s work. For example PMID: 15157838

-          Line 51: This is not entirely true. There are some in vitro (PMID: 9406846, PMID: 15466465, PMID: 27312083, PMID: 28605984) and in vivo studies (PMID: 197120) indicating that T4 can also bind to thyroid hormone receptors, although to a much lesser extent than T3. I understand that it is not the scope of this work to do a thorough revision about this topic, but it should be either briefly mentioned or wording should be chosen carefully to not exclude this possibility.

-          Lines 58 -59. This sentence describes only positive transcriptional regulation by T3, however, negatively transcriptional regulation by T3 is also relevant.

-          Lines 125: “Neuronal tissue” only includes neurons. Neural or brain tissue would include all different neural cell types.

-          Line 142: Figure 3. Why are there double transcripts per million when using Exon Proxy instead of StringTie? A small justification for the non-specialised reader would reinforce that exons can indeed be used as a proxy of the whole transcript.

-          Line 174-177: Please, revise this sentence.

-          Line 190: “human embryos from Carnegie”. I think the manuscript lacks a proper description of this dataset. Is it whole embryo tissue? Only brain tissue? All brain regions?

-          Line 198: a better description of how the different markers recapitulate the different stages of neuronal differentiation would improve the understanding.

-          Lines 214 – 228. Figure 6. A brief description of the different clusters of neurons, NPCs, oRG and RG would provide additional useful information.

Author Response

[1] Line 44: When talking about thyroid hormone replacement treatment shortly after birth to prevent neurological impairments, I strongly recommend also citing Gabriella Morreale’s work. For example PMID: 15157838

Answer: We followed the suggestion of the reviewer and cited the publication of Escobar et al. (2004).

[2] Line 51: This is not entirely true. There are some in vitro (PMID: 9406846, PMID: 15466465, PMID: 27312083, PMID: 28605984) and in vivo studies (PMID: 197120) indicating that T4 can also bind to thyroid hormone receptors, although to a much lesser extent than T3. I understand that it is not the scope of this work to do a thorough revision about this topic, but it should be either briefly mentioned or wording should be chosen carefully to not exclude this possibility.

Answer: We agree with the reviewer and have now rephrased the sentence as follows "THRα is a nuclear hormone receptor for the active hormone triiodothyronine (T3). T4 mainly acts as a prohormone to T3, although some recent evidence has shown weak binding to THRα, but to a much lesser extent than T3 [1]." and cited the paper by Sander et al. 2004

[3] Lines 58 -59. This sentence describes only positive transcriptional regulation by T3, however, negatively transcriptional regulation by T3 is also relevant.

Answer: We agree and have inserted the following passage "In some instances, e.g. for the transcriptional regulation of thyrotropin releasing hormone (TRH), binding of T3 to THRα blocks TRH gene transcription in the hypothalamus [2]." and cited the work by Guissouma et al. (2000).

[4] Lines 125: “Neuronal tissue” only includes neurons. Neural or brain tissue would include all different neural cell types.

Answer: We agree and have changed the word.

[5] Line 142: Figure 3. Why are there double transcripts per million when using Exon Proxy instead of StringTie? A small justification for the non-specialised reader would reinforce that exons can indeed be used as a proxy of the whole transcript.

Answer: We have now added the following explanation to the legend of Figure 3: "The increased TPM values of (b) compared to (a) agree with the fact that TPM calculations normalize against transcript length and hence using the shorter exon instead of the full transcript results in higher TPM values."

[6] Line 174-177: Please, revise this sentence.

Answer: We are sorry that the sentence had been truncated probably due to multiple edits. We have now corrected it. It now reads "To date, there is a lack of publicly available scRNA-seq datasets of human cortical organoids or human embryos generated with a splicing isoform-sensitive platform (e.g. Smart-Seq2 and long-read sequencing)."

[7] Line 190: “human embryos from Carnegie”. I think the manuscript lacks a proper description of this dataset. Is it whole embryo tissue? Only brain tissue? All brain regions?

Answer: We added a sentence about how the authors had processed the embryos: "The original authors had processed the embryos by cutting them and preparing the separate sequencing libraries of each segment. We downloaded FastQ files of the samples annotated by the original authors as either “head_trunk” or “head”."

[8] Line 198: a better description of how the different markers recapitulate the different stages of neuronal differentiation would improve the understanding.

Answer: We agree and have reformulated the passage in the following way: "The identified cell populations recapitulate different stages of cortical development. This includes the progenitor pool of radial glia (marker: VIM) and outer radial glia (markers: FABP7, PTN), a pool of cells positive for neuronal differentiation markers (GAP43, DCX, SYT1, STMN2) and finally pools of cells that were positive for markers of early oligodendrogenesis (PLP1) and early astrocytogenesis (APOE). "

[9] Lines 214 – 228. Figure 6. A brief description of the different clusters of neurons, NPCs, oRG and RG would provide additional useful information.

Answer: We have modified the legend to Figure 6 as follows: "(b) UMAP plot showing the cell types identified by manual curation. These cells include a group of (i) progenitor radial glial cells (RG-1, RG-2, RG-enriched ribosome and proliferating RG), (ii) outer radial glial cells (oRG-1 to oRG-5), (iii) intermediate neuronal precursor cells (NPCs-1 and NPCs-2), (iv) early differentiated neurons (Neurons-1 to Neurons-4 and Neurons-ribosome enriched), (v) early astrocyte precursor cells (APCs), (vi) early oligodendrocyte precursor cells (OPCs), and finally (vii) Choroid plexus TTR+ cells (Choroid plexus cells)."

References

  1. Sandler, B.; Webb, P.; Apriletti, J.W.; Huber, B.R.; Togashi, M.; Lima, S.T.C.; Juric, S.; Nilsson, S.; Wagner, R.; Fletterick, R.J.; et al. Thyroxine-Thyroid Hormone Receptor Interactions*. J. Biol. Chem. 2004, 279, 55801–55808, doi:10.1074/jbc.M410124200.
  2. Guissouma, H.; Becker, N.; Seugnet, I.; Demeneix, B.A. Transcriptional Repression of TRH Promoter Function by T3: Analysis by in Vivo Gene Transfer. Biochem. Cell Biol. Biochim. Biol. Cell. 2000, 78, 155–163.
  3. Tagami, T.; Kopp, P.; Johnson, W.; Arseven, O.K.; Jameson, J.L. The Thyroid Hormone Receptor Variant Α2 Is a Weak Antagonist Because It Is Deficient in Interactions with Nuclear Receptor Corepressors*. Endocrinology 1998, 139, 2535–2544, doi:10.1210/endo.139.5.6011.
  4. Paisdzior, S.; Knierim, E.; Kleinau, G.; Biebermann, H.; Krude, H.; Straussberg, R.; Schuelke, M. A New Mechanism in THRA Resistance: The First Disease-Associated Variant Leading to an Increased Inhibitory Function of THRA2. Int. J. Mol. Sci. 2021, 22, 5338, doi:10.3390/ijms22105338.
  5. Souza, P.C.T.; Puhl, A.C.; Martínez, L.; Aparício, R.; Nascimento, A.S.; Figueira, A.C.M.; Nguyen, P.; Webb, P.; Skaf, M.S.; Polikarpov, I. Identification of a New Hormone-Binding Site on the Surface of Thyroid Hormone Receptor. Mol. Endocrinol. 2014, 28, 534–545, doi:10.1210/me.2013-1359.

Reviewer 2 Report

Comments and Suggestions for Authors

In this paper, the authors examined the expression levels of THRA1 and THRA2 in the brain using RNA-seq data. This study provides novel insights, and the findings will be valuable to researchers in the fields of developmental neurology, endocrinology, and nuclear receptors. The paper is well-written, and I have only a few minor comments.

・       According to the protein structure, Exon 9b in THRA1 gene code Helix10-11 which is well known as interface for homodimer or heterodimer of nuclear receptor. Thus, the illustration in Fig1b is questionable if THRα2 which lacks Exon 9b forms homo or heterodimer. 

・       In addition, the reviewer thinks that THRα generally forms heterodimer with RXRα for transcription of target gene, but Fig 2 shows they form homodimer. 

・       Furthermore, the reviewer believes that only one T3 molecule can access the ligand-binding pocket in THRα, but the illustration shows two T3 molecules binding to a single THRα.

・       It is important to understand the transcriptional activity of THRα1 in various tissues, as shown in Figure 2, where THRA1 and THRA2 levels differ. The reviewer recommends including the expression levels of target genes.

Author Response

[1] According to the protein structure, Exon 9b in THRA1 gene code Helix10-11 which is well known as interface for homodimer or heterodimer of nuclear receptor. Thus, the illustration in Fig1b is questionable if THRα2 which lacks Exon 9b forms homo or heterodimer.

Answer: We agree with the reviewer that the experimental evidence is weak about the possible heterodimerization between THRα1 and THRα2 and its potential physiological effect. We have here referred to the in vitro experimental data provided by Tagami et al. (1998), Table 1 [3]. The fact whether homo- or heterodimers (THRα1|THRα2, THRα1|RXRα, THRα2|RXRα) occur or not, seems not only to be dependent on the proteins, but also on the respective DNA-binding motifs, i.e., the thyroid hormone response elements (TREs). We did not find a crystal structure of any of the THRα1|THRα2, THRα1|RXRα, and THRα2|RXRα heterodimers that would inform us about the exact interface surface of the proteins. We have therefore modified Figure 1b to depict only firmly established knowledge (THRα1|RXRα; THRα1|THRα1 and THRα2|RXRα).

[2] In addition, the reviewer thinks that THRα1 generally forms heterodimer with RXRα for transcription of target gene, but Fig 2 shows they form homodimer.

Answer: We agree with the notion that THRα1 forms a heterodimer with RXRα AND a homodimer with itself and have changed the Figure 1b accordingly. Here we again refer to the in vitro data of Tagami et al. (1998) (Figure 1A) and to our own functional data by Paisdzior et al. (2021) [4] that THRα1 indeed forms homodimers.

[3] Furthermore, the reviewer believes that only one T3 molecule can access the ligand-binding pocket in THRα, but the illustration shows two T3 molecules binding to a single THRα.

Answer: We had referred to the publication of Souza et al. (2014) [5], who found in X-ray crystallographic investigations two T3 binding pockets, one in the center and one in the periphery with different binding strengths and configurations. In the crystallographic structures, both sites were occupied by thyroid hormone(s). However it is presently unknown whether this second binding pouch is active and relevant in normal physiology and whether one or two T3 molecules might bind in vivo. Since the discussion of this finding and its physiology is not the main direction of our article, we have changed Figure 1b back to a single T3 binding site.

[4] It is important to understand the transcriptional activity of THRα1 in various tissues, as shown in Figure 2, where THRA1 and THRA2 levels differ. The reviewer recommends including the expression levels of target genes.

Answer: We thank the reviewer for his/her suggestion and have included the expression levels of some well-known T3-dependent genes (e.g. RXRA, IGF1, and DIO3) in Figure S1. We now write in the text: "In order to look for a potential functional relevance of different THRA1:THRA2 ratios, we provided the TPM values of the well-known THRA target genes IGF1, DIO3, and RXRA in Figure S1, where we see a tendency for higher expression in tissues with a comparatively lower THRA2 expression. However, as a caveat, the levels of thyroid hormone receptor (THR) regulated genes in different tissues are likely to be determined by many factors other than the splicing ratio of THRA isoforms. The final expression levels of target genes would therefore depend on factors such as the amount of T3 generated in these cells (e.g., by the various deiodinases), the co-presence of THRB and, importantly, on epigenomic factors that dictate promoter accessibility for any given THR regulated gene. In addition, many THR regulated genes are tissue specific in the first place, which would make direct comparisons very challenging."

References

  1. Sandler, B.; Webb, P.; Apriletti, J.W.; Huber, B.R.; Togashi, M.; Lima, S.T.C.; Juric, S.; Nilsson, S.; Wagner, R.; Fletterick, R.J.; et al. Thyroxine-Thyroid Hormone Receptor Interactions*. J. Biol. Chem. 2004, 279, 55801–55808, doi:10.1074/jbc.M410124200.
  2. Guissouma, H.; Becker, N.; Seugnet, I.; Demeneix, B.A. Transcriptional Repression of TRH Promoter Function by T3: Analysis by in Vivo Gene Transfer. Biochem. Cell Biol. Biochim. Biol. Cell. 2000, 78, 155–163.
  3. Tagami, T.; Kopp, P.; Johnson, W.; Arseven, O.K.; Jameson, J.L. The Thyroid Hormone Receptor Variant Α2 Is a Weak Antagonist Because It Is Deficient in Interactions with Nuclear Receptor Corepressors*. Endocrinology 1998, 139, 2535–2544, doi:10.1210/endo.139.5.6011.
  4. Paisdzior, S.; Knierim, E.; Kleinau, G.; Biebermann, H.; Krude, H.; Straussberg, R.; Schuelke, M. A New Mechanism in THRA Resistance: The First Disease-Associated Variant Leading to an Increased Inhibitory Function of THRA2. Int. J. Mol. Sci. 2021, 22, 5338, doi:10.3390/ijms22105338.S
  5. Souza, P.C.T.; Puhl, A.C.; Martínez, L.; Aparício, R.; Nascimento, A.S.; Figueira, A.C.M.; Nguyen, P.; Webb, P.; Skaf, M.S.; Polikarpov, I. Identification of a New Hormone-Binding Site on the Surface of Thyroid Hormone Receptor. Mol. Endocrinol. 2014, 28, 534–545, doi:10.1210/me.2013-1359.